# VIDEOALCHEMY:
# OPEN-SET PERSONALIZATION IN VIDEO GENERATION

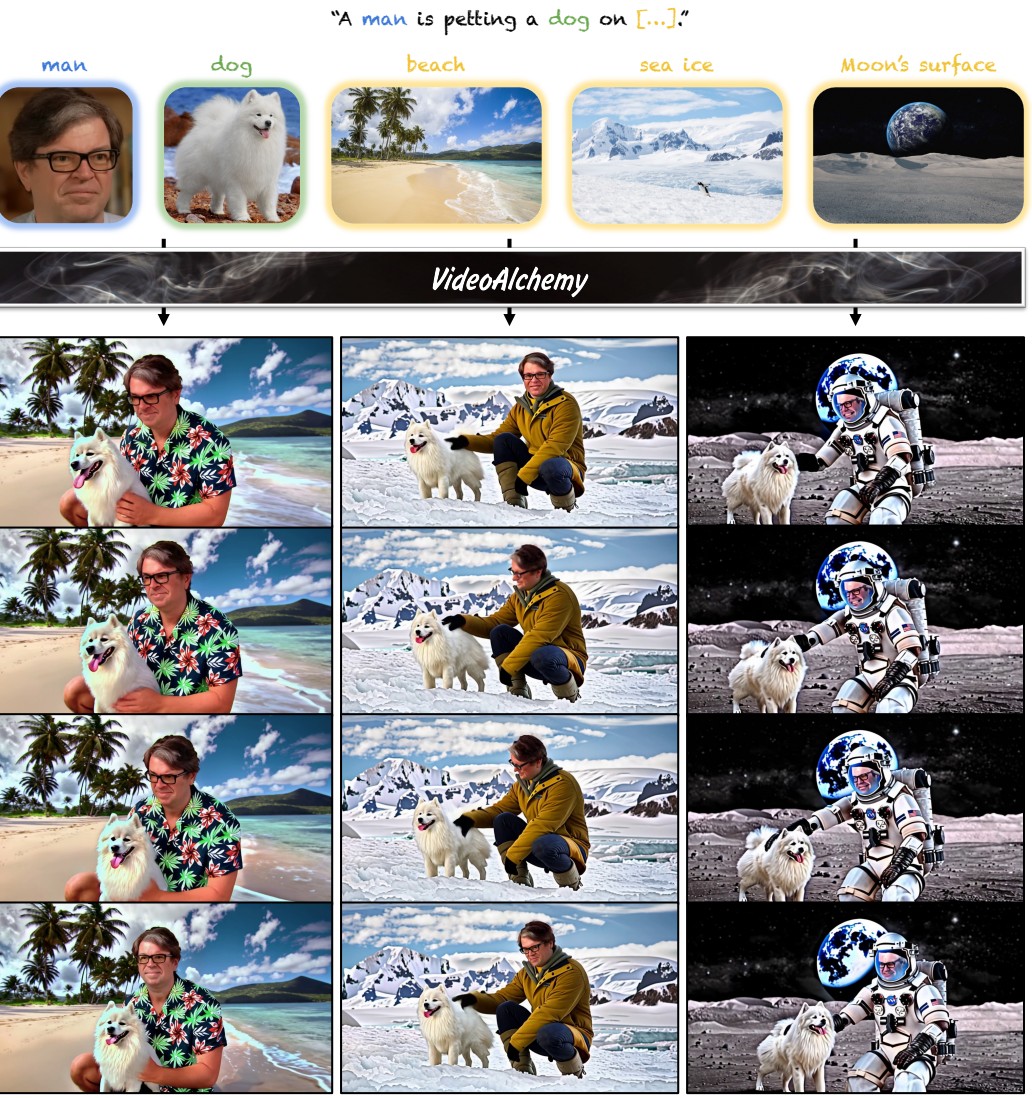

Figure 1: **Overview.** Given a text prompt as well as reference images for each subject (man, dog) and background images (beach, sea ice, moon's surface), *VideoAlchemy* is able to synthesize natural motions while preserving subject identity and background fidelity.

## ABSTRACT

Video personalization methods allow us to synthesize videos with specific concepts such as people, pets, and places. However, existing methods often focus on limited domains, require time-consuming optimization per subject, or support only a single subject. We present *VideoAlchemy*—a video model equipped with built-in multi-subject, open-set personalization capabilities for both foreground objects and backgrounds, eliminating the need for time-consuming test-time optimization. Our model is built on a new Diffusion Transformer module that fuses

each reference image conditioning and its corresponding subject-level text prompt with cross-attention layers. Developing such a large model presents two main challenges: *dataset* and *evaluation*. First, as paired datasets of reference images and videos are extremely hard to collect, we opt to sample video frames as reference images and synthesize entire videos. This approach, however, introduces data biases issue, where models can easily denoise training videos but fail to generalize to new contexts during inference. To mitigate these issues, we carefully design a new automatic data construction pipeline with extensive image augmentation and sampling techniques. Second, evaluating open-set video personalization is a challenge in itself. To address this, we introduce a new personalization benchmark with evaluation protocols focusing on accurate subject fidelity assessment and accommodating different types of personalization conditioning. Finally, our extensive experiments show that our method significantly outperforms existing personalization methods, regarding quantitative and qualitative evaluations.

# 1 INTRODUCTION

Diffusion models (Ho et al., 2020; Sohl-Dickstein et al., 2015; Song & Ermon, 2019) have enabled us to synthesize realistic videos with natural motions given a simple text prompt (Singer et al., 2023; Blattmann et al., 2023b; Brooks et al., 2024; Ho et al., 2022; Menapace et al., 2024). This level of quality and realism paves the way for personalization—the ability to generate videos containing specific objects and people rendered in the unseen context or background. Multiple video personalization methods have been proposed to generate content with specific people or pets, but they remain limited in the level of control they provide. Some focus on human faces (He et al., 2024; Ma et al., 2024), some support only a single subject (Jiang et al., 2024; Wei et al., 2024; Zhou et al., 2024; Wu et al., 2024), with others supporting only foreground control (Wang et al., 2024c). Moreover, some of these works require costly and lengthy test-time optimization (Wei et al., 2024; Wu et al., 2024).

In this paper, we present *VideoAlchemy*, a video generation model with extensive personalization capabilities. In contrast to existing methods, *VideoAlchemy* supports multiple subjects and open-set entities, including both foreground objects and background. Importantly, our optimization-free method does not require fine-tuning to incorporate new concepts. In Figure 1, we show videos personalized for two subjects across three different backgrounds. Our video model is built on new Diffusion Transformer modules tailored for personalization. Each module uses two cross-attention layers: one to integrate the text prompt describing the entire image and another to incorporate the embeddings of each reference image. We employ object-level fusion, blending the text description of each object with its corresponding image embeddings to achieve multiple subject conditioning.

But how can we collect data to train our model? Ideally, it requires a dataset of images and videos with many subjects, each captured under varying lighting, background, and pose. Unfortunately, collecting such a dataset for open-set entities is challenging at best and impossible at worst. Alternatively, we can extract the reference images and target video clips from the same video. This approach, however, comes with a significant drawback—factors unrelated to identity still have a very high correlation across different video frames. While this correlation helps the model denoise training videos accurately, the model often struggles to synthesize diverse videos with unseen lighting, background, and poses. To address these biases, we carefully design a data construction pipeline to automatically extract object segments from target videos. Additionally, we craft a personalization-specific data augmentation and conditional subject sampling strategy during training to ensure the model focuses on the object identity of the reference images.

Another challenge we are facing is the lack of a suitable benchmark for evaluating multi-subject video personalization. Commonly, we evaluate video personalization results by computing a similarity score between the generated video and reference images (Ruiz et al., 2023a; Ye et al., 2023; Jiang et al., 2024; Zhou et al., 2024). Unfortunately, this metric does not apply to multiple entities, as it cannot focus on each subject. To address these limitations, we introduce *MSRVTT-Personalization*, a comprehensive and robust evaluation protocol for personalization tasks. *MSRVTT-Personalization* facilitates evaluation across various conditioning modes, including face-crop conditioning, conditioning on single or multiple arbitrary subjects, and conditioning on foreground and background. Different from image-level similarity, we use object segmentation algorithm to localize each con-

cept in the generated video frames. The experiments demonstrate that our method outperforms existing personalization methods in terms of both quantitative and qualitative assessments. The main contributions of this paper are summarized as follows:

- We present *VideoAlchemy*, a new video generation model, supporting multi-subject, open-set personalization capabilities for both foreground and background.
- We carefully curate a large-scale training dataset and introduce training techniques to prevent the model from learning unintended data biases.
- We introduce *MSRVTT-Personalization*, a new benchmark for the task of personalization, providing various conditioning modes and accurate measurement of subject fidelity.

## 2 RELATED WORK

**Diffusions Model for Video Generation.** Diffusion models (Sohl-Dickstein et al., 2015; Song & Ermon, 2019; Ho et al., 2020; Rombach et al., 2022; Ho et al., 2022) have demonstrated impressive capabilities in generating realistic content in recent years. Building on the power of diffusion models, several subsequent studies have explored their applications in text-conditioned video synthesis (Saharia et al., 2022; Singer et al., 2023; Blattmann et al., 2023b; Zhou et al., 2022; Luo et al., 2023; Guo et al., 2024; Menapace et al., 2024; Brooks et al., 2024). ImagenVideo (Saharia et al., 2022) and Make-A-Video (Singer et al., 2023) propose a cascade of temporal and spatial upsamplers for video generation. VideoLDM (Blattmann et al., 2023b) adopts a latent diffusion paradigm where a pretrained latent image generator and latent decoder are finetuned to generate temporally coherent videos. Differently from previous models based on the U-Net (Ronneberger et al., 2015) architecture, SnapVideo (Menapace et al., 2024) adapts the FiT (Chen & Li, 2023) and scales up to billion-parameters size. More recently, SORA (Brooks et al., 2024) employs the Diffusion Transformer (DiT) (Peebles & Xie, 2023) and shows a tremendous leap in high-resolution, long video synthesis. While these studies demonstrate significant advancements in video synthesis, the use of text prompts alone confines the generated content to what can be described textually.

**Personalized Image Generation.** This task aims at adapting and customizing generative models to a set of desired subjects using a few input images (Ruiz et al., 2023a; Gal et al., 2023a; Kumari et al., 2023; Ye et al., 2023; Shi et al., 2024; Tewel et al., 2024; Wang et al., 2024b; Ostashev et al., 2024). For example, DreamBooth (Ruiz et al., 2023a) optimizes the weights of the entire text-to-image model for a reference subject. Textual Inversion (Gal et al., 2023a) learns a text embedding of the reference subject and uses the embedding to generate novel images. Custom Diffusion (Kumari et al., 2023) learn to compose multiple concepts, each represented by the text embedding and cross-attention weights. However, these optimization-based models require finetuning pre-trained weights or optimizing a text embedding for every new concept, which is inevitably slow and prone to overfitting. Recently, more studies have explored encoder-based methods to reduce test-time finetuning (Shi et al., 2024; Ye et al., 2023; Arar et al., 2023; Gal et al., 2023b; Wei et al., 2023b; Li et al., 2023; Valevski et al., 2023; Ruiz et al., 2023b). IP-adapter (Ye et al., 2023) learns a lightweight decoupled cross-attention mechanism for image conditioning. InstanceBooth (Shi et al., 2024) trains an image encoder to convert reference images into textual tokens and introduces adapter layers to retain identity details. Our model also trains an image encoder for faster personalization, but we focus on video personalization with multiple subjects.

**Personalized Video Generation.** Inspired by the success in image personalization, several works have explored these techniques for videos (Zhang et al., 2024; Jiang et al., 2024; Wei et al., 2024; Wang et al., 2024c; Long et al., 2024; Zhou et al., 2024; Wu et al., 2024; He et al., 2024; Fang et al., 2024). Among them, DreamVideo (Wei et al., 2024) employs an optimization-based strategy, training an image adapter to capture the subject's appearance and a motion adapter to model dynamics. StoryDiffusion (Zhou et al., 2024) instead adopts an optimization-free approach by introducing a consistent self-attention mechanism and employing a semantic motion predictor to synthesize videos with smooth transitions and consistent subjects. Nonetheless, most existing video personalization methods focus on limited domains. Some models are limited to face personalization (He et al., 2024; Ma et al., 2024) or single subjects from specific categories (Zhang et al., 2024; Jiang et al., 2024; Wei et al., 2024; Zhou et al., 2024; Wu et al., 2024), while the other focuses solely on foreground objects (Wang et al., 2024c). In contrast, our work introduces a video model with extensive personalization capabilities, supporting the customization of multiple open-set entities across both foreground and background. Closely related to our work, VideoDrafter (Long et al., 2024) achieves

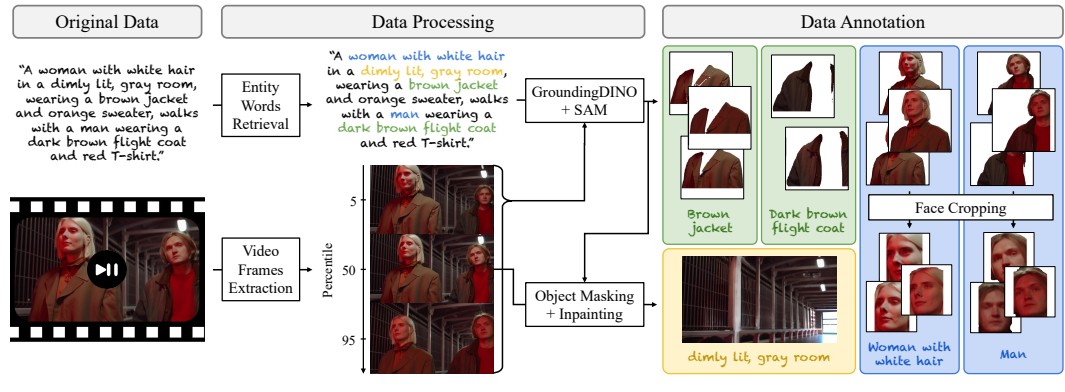

Figure 2: **Dataset collection pipeline for video personalization.** We construct our training dataset using video and caption pairs through a three-step process. First, we identify three categories of entity words from the captions: subject, object, and background. Next, we use the identified object entity words to localize and segment the target subject within three selected video frames. Finally, we extract the clean background by removing the subjects and objects from the middle frame.

open-set video personalization in two stages: image personalization and animation. In contrast, our end-to-end method avoids the issue of poor subject consistency in long video synthesis, a notable limitation of first-frame animation methods.

## 3 METHODOLOGY

Given a text prompt and a set of images conceptualizing each entity word in the prompt, our goal is to learn a video generative model conditional on both text and image inputs. We first elaborate on the collection of the training dataset in Section 3.1, and provide the details of the model architecture in Section 3.2. Lastly, we discuss the issue of training data biases and our solution in Section 3.3.

### 3.1 DATASET COLLECTION

As shown in Figure 2, we curate the training dataset upon video and caption pairs with three steps. In the first step, we use a large language model (Jiang et al., 2023) to retrieve entity words from the given caption. Specifically, we define three types of entity words: subjects (*e.g.*, human or animal), objects (*e.g.*, car, jacket), and backgrounds (*e.g.*, room, beach). Each subject or object entity word is expected to appear in the video. Next, we use the retrieved entity words to filter the training dataset with the following criteria: (1) we remove videos containing any subject entity word in plural form (*e.g.*, a group of people, multiple dogs), as they introduce ambiguity in model personalization; (2) we also remove videos without any subject entity words, as their dynamics are often dominated by camera movements rather than significant foreground motion. Appendix A.2 details this process.

In the second step, we construct reference images that feature subjects and objects. We first select three frames from the video's beginning, middle, and end (at the 5%, 50%, and 95% percentiles), which might capture the target subject or object with varying poses and different lighting conditions. Next, we apply GroundingDINO (Liu et al., 2023a) on each frame to detect the bounding boxes. These bounding boxes are then used by SAM (Kirillov et al., 2023) to segment the mask regions corresponding to each entity. Additionally, for the reference images that depict humans, we apply face detection (Wang et al., 2024a) to extract face crops.

Lastly, we create a clean background image by removing the subjects and objects. Since SAM (Kirillov et al., 2023) occasionally produces imprecise boundaries, we dilate the foreground mask. Next, we use an inpainting algorithm (Rombach et al., 2022) to obtain a clean background image. We use the background entity word as the positive prompt and "*Any human or any object, complex pattern and texture*" as the negative prompt. To ensure consistency of the background, we only use the middle frame to obtain a single background image for each video sequence.

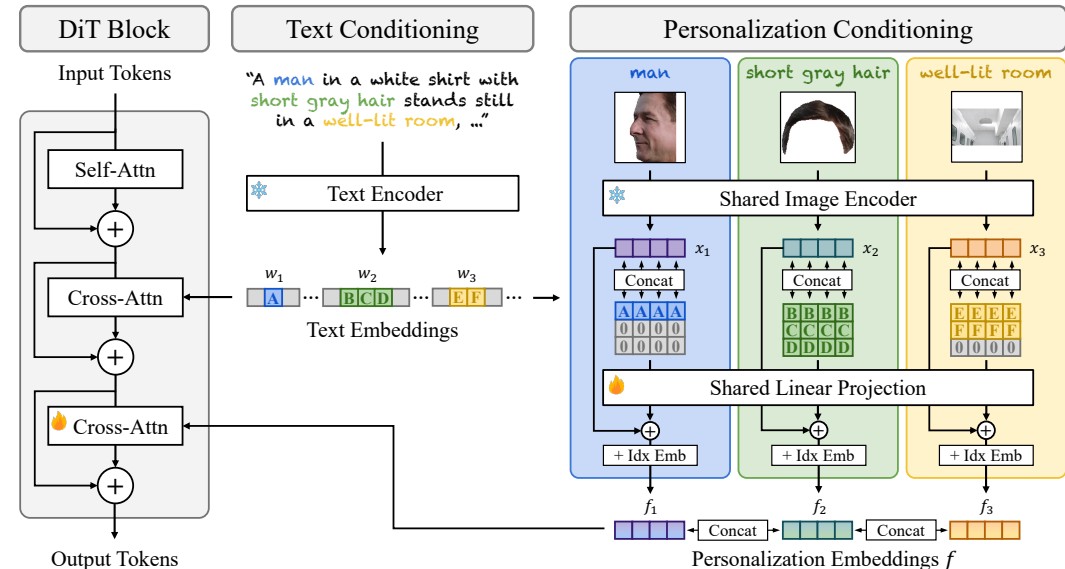

Figure 3: **Model architecture.** We use the DiT (Peebles & Xie, 2023) architecture, consisting of a cascade of DiT blocks, as the backbone of video synthesis. In each DiT block, we perform an additional cross-attention operation with personalization embeddings, which encompass information from both the image and its representative entity word. Each square in the figure is a 1-D token.

## 3.2 VIDEO PERSONALIZATION MODEL

In Section 3.1, we annotate each video and prompt pair with a sequence of reference images and their corresponding entity words. Next, we train *VideoAlchemy* by learning to denoise the training video using the conditions of text prompt, reference images, and conditional entity words. Figure 3 illustrates the model architecture of *VideoAlchemy*, a deep cascade of Diffusion Transformer (DiT) blocks (Peebles & Xie, 2023). Different from vanilla DiT designs, our module supports personalization by fusing the information from both text and image conditioning. Our DiT block includes three main operations: one multi-head self-attention (Vaswani, 2017) and two following multi-head cross-attention respectively for text and reference image conditioning.

**Binding of Image and Word Concept.** In the task of multi-subject, open-set personalization, the video model can be conditioned on different subjects, each of which can be represented by one or a few reference images. Therefore, it is critical to provide the model binding information between text tokens and image tokens. We provide these binding in the form of personalization embeddings $f = \mathrm{Concat}(f_1, ..., f_N)$, where $f_n$ encompasses the information from both the reference image and the representative entity word and $N$ is the number of conditional reference images. Specifically, to produce the embeddings $f_n$, we first encode the image as the image tokens $x_n \in \mathcal{R}^{l \times c}$ by a shared and frozen image encoder. Next, we retrieve the word tokens $w_n$ from the text embeddings (encoded from the text prompt), and flatten $w_n$ to a 1-D embedding. Considering the number of tokens of an entity word varies, we zero-pad or crop the word embeddings to a consistent length. To bind the information of both the image and word tokens, we repeat the flattened word tokens for $l$ times and concatenate them with the image tokens along the channel axis. Lastly, after a linear projection module, we apply a residual connection with the image tokens $x_n$ and add a learnable index embedding to produce the embeddings $f_n$.

**Personalization Conditioning.** The personalization embeddings $f$ are later used to compute cross attention with video latent tokens. Note that IP-adapter (Ye et al., 2023) encodes conditional text and image into an unified embeddings space through CLIP (Radford et al., 2021) and employs a single decoupled cross-attention to compute both conditioning at the same time. In contrast, our model encodes the text and image using separate models. Empirically, we find that using distinct cross-attention for each modality can handle the tokens from different distributions more effectively.

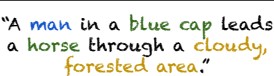
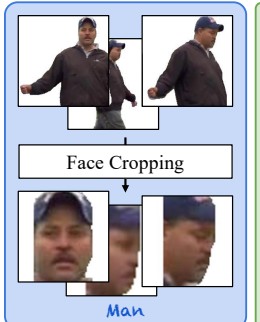
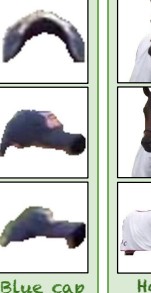
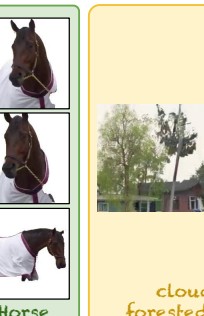

Figure 4: **Test sample in *MSRVTT-Personalization* benchmark.** We present a comprehensive benchmark for personalization models. By sampling different reference images as inputs, our benchmark supports various conditioning modes: face conditioning, single or multiple arbitrary subjects conditioning, and both foreground and background conditioning.

### 3.3 Undesirable Training Data Biases

In Section 3.2, we learn *VideoAlchemy* by denoising the entire video from the selected and masked video frames. Empirically, we observe that this training strategy leads the model to learn unintended biases presented in the reference image (*ref*). We list some noticeable biases as follows:

- If *ref* is high-resolution, the model generates a large object close to the camera.
- If *ref* has been photoshopped, the model replicates the subject without introducing motion.
- If *ref* is occluded, the model generates random objects occluding the subject.
- If *ref* is cropped, the model places the subject at edge to make it cropped by the boundary.
- The model tends to generate the subject with the same pose and lighting conditions as *ref*.
- If multiple *ref*s represent the same subject with similar poses, the model generates a subject with small motion.

During training, our model learns to exploit these biases since they are beneficial at denoising training video. Nonetheless, they are not applicable during inference. Such domain gap between training and inference usually results to unnatural composition of the objects or undesirable video dynamics. To alleviate these unfavorable biases, we apply a sampling rule to randomly select reference images for conditioning and adopts data augmentations on the reference images. Specifically, we add downscaling and Gaussian blurring to fix the bias on image resolution, apply color jittering and brightness adjustment to mitigate the bias on lighting condition, and adopt random horizontal flip and image shearing and rotation to alleviate the bias on the pose of reference subject.

The core concept is to guide the model focusing on the identity of the reference images instead of learning the unintended information leakage from the properties or composition of the input reference images. We detail the training augmentations and the sampling of the conditioning subjects and images in Appendix A.3.

## 4 Experiments

In Section 4.1, we introduce *MSRVTT-Personalization*, a comprehensive benchmark for personalization. We provide quantitative and qualitative evaluations in Section 4.2 and Section 4.3, respectively. Appendix A contains the details of the training dataset and Appendix B includes the details of model architecture, training, and inference.

### 4.1 MSRVTT-Personalization Benchmark

Existing personalization frameworks (Ruiz et al., 2023a; Ye et al., 2023; Wei et al., 2024; Zhou et al., 2024) assess the preservation of the subject appearance by measuring the similarity (Deng et al., 2019; Radford et al., 2021; Oquab et al., 2024) between the reference image and the entire output image or video frames. However, these metrics are limited to single-subject conditioning and fail to focus specifically on the target subject. To solve this issue, we present *MSRVTT-Personalization*, a

framework designed to provide a more comprehensive and accurate evaluation for the tasks of personalization. It supports various conditioning scenarios: face-crop conditioning, single or multiple arbitrary subject conditioning, and both foreground and background conditioning.

We construct the testing dataset upon MSR-VTT (Xu et al., 2016) and process the dataset with three steps. First, we use TransNetV2 (Souček & Lokoč, 2020) to split a long video into multiple clips based on shot boundary detection and apply the in-house captioning algorithm to generate more a detailed caption for each clip. In the second step, we follow Section 3.1 to produce the annotations for each video-caption pair. Lastly, to ensure the quality of the benchmark, we manually select the samples that meet the following four criteria:

- The video sample is not an animation of an image without any object motion.
- The video sample does not include extensive texts.
- The retrieved subjects and objects cover all of the main subjects and objects in the video.
- Inpainting of the background image does not introduce any additional random objects.

To increase the data diversity, we select only one clip from each long video and collect $2,130$ clips in total, forming the testing samples of the benchmark. Figure 4 shows a test sample with its annotation. To perform an extensive evaluation, we compute four metrics:

- Text-Sim: the average cos-sim between the text prompt and the synthetic video frames.
- Video-Sim: the pairwise cos-sim between the target and the synthetic video frames.
- Subject-Sim: the pairwise cos-sim between the input reference images and the synthetic subject image segmented from the video frames.
- Face-Sim: the pairwise cos-sim between the input face crops and the synthetic face images cropped from the video frames.
  (cos-sim stands for cosine similarity)

With more details, we follow the default setting in Torchmetrics (2024) and use CLIP ViT-L/14 (Radford et al., 2021) embeddings for the Text-Sim and Video-Sim. For the Subject-Sim, we follow Ruiz et al. (2023a) and Wei et al. (2024) and use DINO ViT-B/16 (Caron et al., 2021) embeddings for the evaluation. For the Face-Sim, we use ArcFace R100 (Deng et al., 2019) embeddings to better extract identity features than general image encoders. To detect the target subjects from the synthetic video frames, we utilize Grounding-DINO Swin-T (Liu et al., 2023a) with the confidence score threshold of $0.4$. To detect the synthetic face crops, we employ YOLOv9-C (Wang et al., 2024a) with the confidence score threshold of $0.2$ and the IoU score threshold of $0.4$. For the video frames with missing subjects or face crops, we assign a similarity score of $0$. The testing dataset and the evaluation protocol will be made publicly available and can serve as a comprehensive personalization benchmark in the future.

## 4.2 QUANTITATIVE EVALUATION

In this section, we quantitatively evaluate *VideoAlchemy* and compare it with the state-of-the-art personalization frameworks on *MSRVTT-Personalization*.

**Experimental Setup.** Given that various personalization frameworks utilize different types of conditional images as inputs, we develop two modes: the subject mode and the face mode, which respectively use the entire subject images or only the face crops as inputs. For the subject mode, we collect $1,736$ testing videos that have exactly one subject in the video and compare them with ELITE (Wei et al., 2023a) and VideoBooth (Jiang et al., 2024). For the face mode, we collect $1,285$ testing videos that have exactly one subject containing face crops and compare them with IP-Adapter-FaceID+ (Ye et al., 2023) and PhotoMaker (Li et al., 2024).

For image personalization models, including ELITE (Wei et al., 2023a), IP-Adapter-FaceID+ (Ye et al., 2023), and PhotoMaker (Li et al., 2024), we use StableVideoDiffusion (Blattmann et al., 2023a) Img2Vid-XT-1-1 to animate the output images as videos. Since most frameworks only support single-image input, we randomly sample one subject image or face crop for conditioning. For PhotoMaker (Li et al., 2024), as an exception, we also provide the results using all available face crops for conditioning. Additionally, we report the results of our model with the inclusion of background conditioning.

Table 1: **Quantitative comparison on subject mode of *MSRVTT-Personalization*.** We highlight the top two models using single or multiple subject images as the condition respectively. [†]We treat output images as single-frame videos for the image personalization model.

| Method | Test-time Optimization | Cond. Images | | Text-Sim↑ | Video-Sim↑ | Subject-Sim↑ |
| --- | --- | --- | --- | --- | --- | --- |
| | | Subject | Background | | | |
| ELITE[†] (Wei et al., 2023a) | ✗ | single | ✗ | 0.2454 | 0.6198 | 0.3593 |
| VideoBooth (Jiang et al., 2024) | ✗ | single | ✗ | 0.2216 | 0.6125 | 0.3954 |
| DreamVideo (Wei et al., 2024) | ✓ | single | ✗ | 0.0000 | 0.0000 | 0.0000 |
| *VideoAlchemy* (with CLIP) | ✗ | single | ✗ | 0.2813 | 0.6813 | 0.4991 |
| *VideoAlchemy* (with DINOv2) | ✗ | single | ✗ | 0.2799 | 0.6902 | 0.5373 |
| DreamVideo (Wei et al., 2024) | ✓ | multiple | ✗ | 0.0000 | 0.0000 | 0.0000 |
| *VideoAlchemy* (with DINOv2) | ✗ | multiple | ✗ | 0.2788 | 0.6986 | 0.5502 |
| *VideoAlchemy* (with DINOv2) | ✗ | multiple | ✓ | 0.2731 | 0.7408 | 0.5446 |

Table 2: **Quantitative comparison on face mode of *MSRVTT-Personalization*.** We highlight the top two models using single or multiple face crops as the condition respectively. [†]We treat output images as single-frame videos for the image personalization models.

| Method | Test-time Optimization | Cond. Images | Text-Sim↑ | Video-Sim↑ | Face-Sim↑ |
| --- | --- | --- | --- | --- | --- |
| | | Face crop | | | |
| IP-Adapter[†] (Ye et al., 2023) | ✗ | single | 0.2513 | 0.6481 | 0.2689 |
| PhotoMaker[†] (Li et al., 2024) | ✗ | single | 0.2776 | 0.5687 | 0.1893 |
| Magic-Me (Ma et al., 2024) | ✓ | single | 0.0000 | 0.0000 | 0.0000 |
| *VideoAlchemy* (with CLIP) | ✗ | single | 0.2830 | 0.6441 | 0.2163 |
| *VideoAlchemy* (with DINOv2) | ✗ | single | 0.2819 | 0.6588 | 0.2852 |
| PhotoMaker[†] (Li et al., 2024) | ✗ | multiple | 0.2751 | 0.5824 | 0.2159 |
| Magic-Me (Ma et al., 2024) | ✓ | multiple | 0.0000 | 0.0000 | 0.0000 |
| *VideoAlchemy* (with DINOv2) | ✗ | multiple | 0.2825 | 0.6658 | 0.3125 |

Table 3: **User preference on subject mode and face mode of *MSRVTT-Personalization*.**

| Method | Preference Ratio↑ | | Method | Preference Ratio↑ | |
| --- | --- | --- | --- | --- | --- |
| | Quality | Fidelity | | Quality | Fidelity |
| ELITE (Wei et al., 2023a) | 0.007 | 0.050 | IP-Adapter (Ye et al., 2023) | 0.038 | 0.239 |
| VideoBooth (Jiang et al., 2024) | 0.017 | 0.061 | PhotoMaker (Li et al., 2024) | 0.236 | 0.114 |
| DreamVideo (Wei et al., 2024) | 0.000 | 0.000 | Magic-Me (Ma et al., 2024) | 0.000 | 0.000 |
| *VideoAlchemy* (with CLIP) | 0.540 | 0.368 | *VideoAlchemy* (with CLIP) | 0.310 | 0.274 |
| *VideoAlchemy* (with DINOv2) | 0.436 | 0.521 | *VideoAlchemy* (with DINOv2) | 0.416 | 0.372 |

We implement and evaluate our models with two different image encoders: CLIP (Radford et al., 2021) and DINOv2 (Oquab et al., 2024). The evaluation results are presented in Table 1 and Table 2, respectively, for the subject mode and face mode.

**Comparison with the State-of-the-Arts.** Our framework significantly outperforms the existing open-set personalization models (Wei et al., 2023a; Jiang et al., 2024) regarding Video-Sim and Subject-Sim scores. Notably, our open-set model can achieve a higher Face-Sim score compared to the other frameworks focused on the face domain (Ye et al., 2023; Li et al., 2024). Additionally, our model achieves higher Subject-Sim and Face-Sim with more conditioning reference images and reaches a higher Video-Sim with additional background conditioning images, showing the advantage of multiple-image conditioning. We also notice that our model yields a slightly lower Text-Sim compared to PhotoMaker (Li et al., 2024). We attribute this behavior to a trade-off between fidelity and text-video alignment. Empirically, we find that a personalization model excelling in preserving

Figure 5: **Qualitative comparison on subject mode of *MSRVTT-Personalization*.**

subject details is more challenging to generate a fully text-aligned video due to the limited flexibility in video synthesis.

**Human Evaluation.** To complement the evaluation, we conduct a user study to assess quality and fidelity. We randomly select 200 testing samples from each mode. For each sample, we show the conditioning image along with four videos generated by different models to 5 participants. The participants are asked to select the video with the best preservation of subject details and the video with the best visual and motion quality. We evaluate the subject mode and face mode separately and show the numbers in Table 3. The results show that our model surpasses the state-of-the-art framework by a huge gap in terms of both quality and fidelity. We also highlight that the fidelity score reported by humans is positively correlated to Subject-Sim and Face-Sim scores in the proposed *MSRVTT-Personalization*, showing the effectiveness of our evaluation protocol.

### 4.3 QUALITATIVE EVALUATION

We visualize the comparisons on the subject mode and face mode respectively in Figure 5 and Figure 6, where Appendix C.2 includes more comparisons on different conditioning subjects. The video samples can be found in the *webpage msrvtt* folder of the supplementary material.

Our method can produce more photorealistic video samples with better preservation of subject details compared to ELITE (Wei et al., 2023a), VideoBooth (Jiang et al., 2024), and IP-Adapter-FaceID+ (Ye et al., 2023). As shown in Figure 6, PhotoMaker (Li et al., 2024) can generate high-quality and text-aligned images; however, the synthetic face expresses a low fidelity to the reference face crop, which is aligned with the observation from the quantitative evaluation in Section 4.2.

### 4.4 EFFECTS OF IMAGE ENCODERS ON VIDEO PERSONALIZATION

Encoding the reference images by different models can significantly affect the performance of a personalization model. In this work, we implement our model in two versions utilizing two different image encoders: CLIP (Radford et al., 2021) and DINOv2 (Oquab et al., 2024) and analyze their behaviors across three aspects: fidelity, text-video alignment, and visual quality.

Figure 6: **Qualitative comparison on face mode of** *MSRVTT-Personalization*.

First, using DINOv2 (Oquab et al., 2024) to encode the reference images yields significantly higher fidelity, which is consistently demonstrated in Table 1 to 3 and Figures 5 and 6 (see the "*car's grill and front lamp*" in Figure 5 and the "*cap's buckle*" in Figure 6). We hypothesize that DINOv2 learns to minimize the self-supervised training objective (Chen et al., 2020) and capture unique features in an image. Therefore, DINOv2 embeddings retain rich visual details, which helps maintain subject details. Second, the model using CLIP (Radford et al., 2021) achieves better text-video alignment, as shown in both the quantitative and qualitative evaluations (see the "*BBC logo and the license plate*" in Figure 5). We assume that CLIP learns to bridge visual and textual modalities, guiding its embeddings to focus on details typically described in the text prompt. This helps the model generate text-aligned videos. Finally, based on the results in Table 3, the model using CLIP embeddings provides better video quality when conditioned on an entire subject image. In contrast, the model adopting DINOv2 embeddings results in higher video quality when conditioned on a face crop. We speculate that CLIP embeddings may convey more high-level semantic information which can simplify the video synthesis when conditioned on an relatively complex subject image. On the other hand, for a face crop image that contains fewer semantics features, richer detail presented in DINOv2 embeddings can enhance the generation of photorealistic faces.

## 5 CONCLUSION

In this paper, we present a new video personalization model, *VideoAlchemy*, which demonstrates a significant advancement in video personalization by addressing the limitations of existing methods. Our method supports multi-subject, open-set personalization capabilities for both foreground and background without the need for time-consuming test-time optimization. Through our approach to dataset construction and augmentation engineering, we have largely mitigated challenges related to data biases, enabling our model to better generalize to real-world settings. Furthermore, we introduce a comprehensive personalization benchmark, which supports the measurement of subject fidelity under various conditioning and scenarios. We hope that this benchmark could facilitate robust evaluation for varying personalization approaches and settings. Finally, we experimentally validate that *VideoAlchemy* outperforms existing methods in both quantitative and qualitative measures. We believe our findings pave the way for future research in video synthesis and open up new possible applications in entertainment, advertisement, and education.

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

# A   DETAILS OF TRAINING DATASETS AND AUGMENTATIONS

## A.1   TRAINING DATASETS AND UNDESIRABLE SAMPLES FILTERING

Our personalization training dataset is built on Panda-70M (Chen et al., 2024) and other in-house video-caption datasets. However, we observe that the quality of the video samples is noisy and the training dataset contains several data distributions that are not ideal for generation. We classify the undesirable training samples into four categories:

- Still foreground image: sample which is an animation of a static image.
- Slight motion: sample with tiny camera movement and static foreground object.
- Screen-in-screen: sample with an image or video overlaying on a background image or video.
- Computer screen recording: sample which computer screen recording (excluding PC game).

We find that training on this data can make our personalization model generate trivial videos by simply replicating the input reference images and pasting them onto a static background without introducing any motion, especially when there is varying illumination across the reference images. To address this, we train a video classification model to filter out these undesirable samples. Specifically, we randomly sample 40K videos from the training dataset and manually annotate them with class labels to indicate whether the sample is desirable, and if not, which category of undesirability it falls into. Using the labels, we finetune VideoMAE (Tong et al., 2022) for video classification. Moreover, as we target generating videos that are free of shot boundaries, we apply TransNetV2 (Souček & Lokoč, 2020) to detect videos containing shot boundaries. We only retain the desirable and shot-free video samples for training.

## A.2   RETRIEVAL OF ENTITY WORDS FROM THE PROMPT

In Section 3.1, we utilize a large-language model (Jiang et al., 2023) (LLM) to retrieve entity words from the prompt. In more detail, we use the prompt template in Figure 7 as an instruction.

---

*Given an image caption, please retrieve the word tags that indicate background, subject, and visually separable objects.*
[Definition of background] the background spaces that appear in most of the image area.
[Definition of subject] human or animal subjects that appear in the image
[Definition of object] the entities that appear in part of the image and can be visually separated with each other.
*All of the word tags need to strictly follow two rules below:*
1) word tag is a noun without any quantifier.
2) word tag is an exact subset of the caption. Do not modify any characters, word and symbols.
*Here are some examples, follow this format to output the results:*
### Caption: A woman in a mask and coat, with long brown hair, shows a small green-capped bottle to the camera.
### Output: {'background': [''], 'subject': ['woman'], 'object': ['mask', 'coat', 'long brown hair', 'green-capped bottle']}

(More examples)

---

Figure 7: **Prompt template for entity word retrieval.**

Given the video caption, the LLM agent is expected to return a string in the dictionary format, where the values are the list of entity words retrieved from the text prompt. We apply the following steps to process the output:

- Remove the sample if the output string is not in the valid dictionary format.
- Remove the sample if any entity word is not a sub-string of the text prompt.
- Reclassify the entity words according to the pre-defined rules. For example, "cloud" is not a visually separable object that should be classified into a background entity word.
- Remove the sample with no subject entity word, as we observe that the motion of these samples is typically trivial camera movements and lacks meaningful foreground motion.
- Remove the sample with the subject entity word in the plural form, as this will introduce ambiguity when applying the localization algorithm.

To this end, we curate a training dataset comprising 37.8M videos. To illustrate the diversity of conditioning subjects within the dataset, we plot a word cloud of entity words from 10K randomly sampled training videos in Figure 8.

Figure 8: **Word cloud of entity words.** We randomly sample 10K videos from the training dataset and plot the word cloud of the conditioning subject and object entity words.

Table 4: **Training augmentations.** We denote the height and width of the input image as $h$ and $w$.

|  | Probability | Hyperparameters | |
| --- | --- | --- | --- |
| Downscale | 1.0 | Scale | $[112 \,/ \max(h, w), 1.0]$ |
| Gaussian blur | 1.0 | Kernel size (p) | $\max(h, w) \,/\, 50$ |
| Color jitter | 1.0 | Scale | $[-0.05, 0.05]$ |
| Brightness | 1.0 | Scale | $[0.9, 1.1]$ |
| Horizontal flip | 0.5 | - | - |
| Shearing (x-axis) | 1.0 | Value (p) | $[-0.05, 0.05] \times w$ |
| Shearing (y-axis) | 1.0 | Value (p) | $[-0.05, 0.05] \times h$ |
| Rotation | 1.0 | Value (°) | $[-20, 20]$ |
| Random crop | 1.0 | Scale | $[0.67, 1.0]$ |

## A.3 TRAINING AUGMENTATIONS AND CONDITIONAL IMAGES SAMPLING

In Section 3.3, we propose to prevent the model from learning the undesirable training data biases by adding image augmentations and randomly sample the conditional subjects and reference images for training. Table 4 lists the training augmentations and the hyperparameter setting. While augmentations can fix some biases from reference images, empirically, we find that the model can also learn the unintended biases from the composition of reference images. Specifically, if we always use all available reference images as conditions during training, the model can generate the target subject with some properties correlated to the number of reference images (*ref*) during inference. Using the text prompt "*A dog is running*" as an example:

- If having 0 *ref*, the model generates a tiny or heavily occluded dog.
- If having 1 *ref*, the model generates a dog running out of the view of the video.
- If having 3 *ref*s of a similar pose, the model generates a dog running in slow-motion.

To avoid the model learning the biases from the composition of reference images, we apply a special rule to sample the conditional subjects and their reference images during training. It includes the following five steps:

- Randomly sample the number of the conditional subjects from 1 to 3.
- Randomly sample the conditional subjects with replacement.
- For each subject, randomly sample the number of conditional reference images from 1 to 3.
- For each subject, randomly sample the conditional reference images with replacement.
- Randomly including the background conditional with a probability of 50%.

Table 5: **Architecture details of autoencoder and video generation backbone.**

| Autoencoder | MAGVIT |
|---|---|
| Base channels | 16 |
| Channel multiplier | $[1, 4, 16, 32, 64]$ |
| Encoder blocks count | $[1, 1, 2, 8, 8]$ |
| Decoder blocks count | $[4, 4, 4, 4, 4]$ |
| Stride of frame | $[1, 2, 2, 2, 1]$ |
| Stride of h and w | $[2, 2, 2, 2, 1]$ |
| Padding mode | replicate |
| Compression rate | $8 \times 16 \times 16$ |
| Bottleneck channels | 32 |
| Use KL divergence | ✓ |
| Use adaptive norm | ✓ (decoder only) |

| Backbone | DiT |
|---|---|
| Input channels | 32 |
| Patch size | $1 \times 2 \times 2$ |
| Patch channels | 4096 |
| Latent token channels | 4096 |
| Positional embeddings | RoPE |
| DiT blocks count | 32 |
| Attention heads count | 128 |
| Window size | 6144 (center) |
| Use flash attention | ✓ |
| Use fused layer norm | ✓ |
| Use self conditioning | ✓ |
| Self conditioning prob. | 0.9 |
| Conditioning channels | 1024 |
| Conditioning subjects | 6 (stage 2 only) |

Table 6: **Architecture details of image encoders.**

| | CLIP | DINOv2 | MAE |
|---|---|---|---|
| Backbone | ViT-L/14 | ViT-L/14 | ViT-L/16 |
| Selective block | 23 | 24 | 24 |
| Selective tokens | patch | patch | patch |
| Tokens count | 256 | 256 | 196 |
| Tokens channels | 1024 | 1024 | 1024 |

Table 7: **Training hyperparameters.** The right table is the setting of stage II and III training.

| Stage | I | II | III |
|---|---|---|---|
| Steps | 490K | 20K | 50K |
| Warmup steps | 10K | - | 5K |
| Samples seen | 2.42B | 21.5M | 53.7M |
| Optimizer | AdamW | | |
| Learning rate | $1e^{-4}$ | | |
| LR scheduler | constant | | |
| Beta | $[0.9, 0.99]$ | | |
| Weight decay | 0.01 | | |
| Gradient clipping | 0.05 | | |
| Dropout | 0.1 | | |

| # frames | Batchsize (sampling weights) | |
|---|---|---|
| | $512p \times 288p$ | $1024p \times 576p$ |
| 17 | 2,048 (40%) | 512 (40%) |
| 49 | 832 (1.5%) | 192 (2.5%) |
| 73 | 512 (1.5%) | 128 (2.5%) |
| 97 | 448 (1.5%) | 64 (2.5%) |
| 121 | 384 (1.5%) | 64 (2.5%) |
| 145 | 256 (1.5%) | - (0%) |
| 193 | 192 (0.83%) | - (0%) |
| 289 | 128 (0.83%) | - (0%) |
| 385 | 64 (0.83%) | - (0%) |

# B DETAILS OF MODEL ARCHITECTURE AND TRAINING

## B.1 MODEL ARCHITECTURE

Our framework is a latent-based diffusion model, using MAGVIT (Yu et al., 2023) and DiT (Peebles & Xie, 2023) as the autoencoder and the video generation backbone respectively. We detail the hyperparameters of our model architecture in Table 5. To accelerate the model, we utilize the positional embeddings and self-attention in RoPE (Su et al., 2024) and adopt flash attention (Dao et al., 2022) and fused layer normalization (2018). We implement the models with three different image encoders, including CLIP (Radford et al., 2021), DINOv2 (Oquab et al., 2024), MAE (He et al., 2022), where the backbone and other details are listed in Table 6. We find that using the patch tokens as the image embeddings can retain more localized properties of the reference images and result to higher fidelity compared to the class token. Moreover, aligned with the observation

from Liu et al. (2024), we notice that CLIP's patch tokens from the second last transformer block can yield better preservation of the subject details than the ones from the last block.

## B.2 MODEL TRAINING

We present the training details of the model in Table 7. We train the model in three stages. In the first stage, we fix the autoencoder and train the video generation backbone without cross-attention for personalization conditioning for 490K steps with a 10K-step warmup. In the second stage, we introduce the personalization conditioning modules and finetune them while keeping the video generation backbone and image encoder fixed for 20K steps. In the final stage, we finetune both the video generation backbone and the personalization conditioning modules, keeping the image encoder fixed, for 50K steps with a 5K-step warmup. We use the AdamW (Loshchilov, 2017) optimizer with a constant learning rate of $1e^{-4}$. To achieve stable training, we set $\beta = [0.9, 0.99]$, a weight decay of $0.01$, gradient clipping with the value of $0.05$. We randomly drop the text prompt or subject image conditioning with a probability of $10\%$ and set them to zero to support classifier-free guidance (Ho & Salimans, 2022).

To enable the generation of high-resolution and long-duration videos while ensuring efficient model training, we train our model on videos of varying resolutions and lengths. Table 7 lists the batchsizes and sampling weights for the training videos across different resolutions and lengths. The batchsizes are set to balance the training time for each step with different attributes. We apply the fixed framerate of $24$. Our model supports generating videos up to 16 seconds in length at $512\text{p} \times 288\text{p}$ resolution, and up to 5 seconds at $1024\text{p} \times 576\text{p}$ resolution.

We implement our model in PyTorch (Paszke et al., 2019) and perform all experiments on Nvidia 80GB A100 GPUs.

## B.3 MODEL INFERENCE

We utilize a rectified flow sampler (Liu et al., 2023b) with classifier-free guidance (Ho & Salimans, 2022) (CFG) for sampling. The choice of CFG scale can significantly impact the performance of diffusion models. While our model performs best with a CFG scale of $8$ for text conditioning, we find that applying such a high CFG scale for subject image conditioning can cause the model to embed reference images directly into the video, without introducing natural motion and appearance variation. To address this, we apply CFG twice within each sampling step: once for text conditioning with a CFG scale of $8$ and once for subject image conditioning with a scale of $2.5$. We use 128 denoising steps for quantitative evaluations and 256 steps for qualitative visualizations, with the same CFG interval (Kynkäänniemi et al., 2024) of $[0.15, 0.5]$. Additionally, we apply time shifting (Esser et al., 2024; Gao et al., 2024) to align the signal-to-noise ratio (SNR) across different resolutions.

# C  MORE VISUALIZATION RESULTS

## C.1 ABLATION STUDY ON DIFFERENT CONDITIONING IMAGES

In this section, we conduct an ablation study on various conditioning images with the same prompt as in Figure 1. Specifically, we ablate different "*person*" images in Figure 9, "*dog*" images in Figure 10, and background images in Figure 11. The video samples and more thorough ablation study are in the *webpage ablation* folder of the supplementary material.

Figure 9: **Ablation study on the conditioning images of "*person*".** The bottom-most conditioning image is synthesized by DALL·E 3 (Betker et al., 2023)

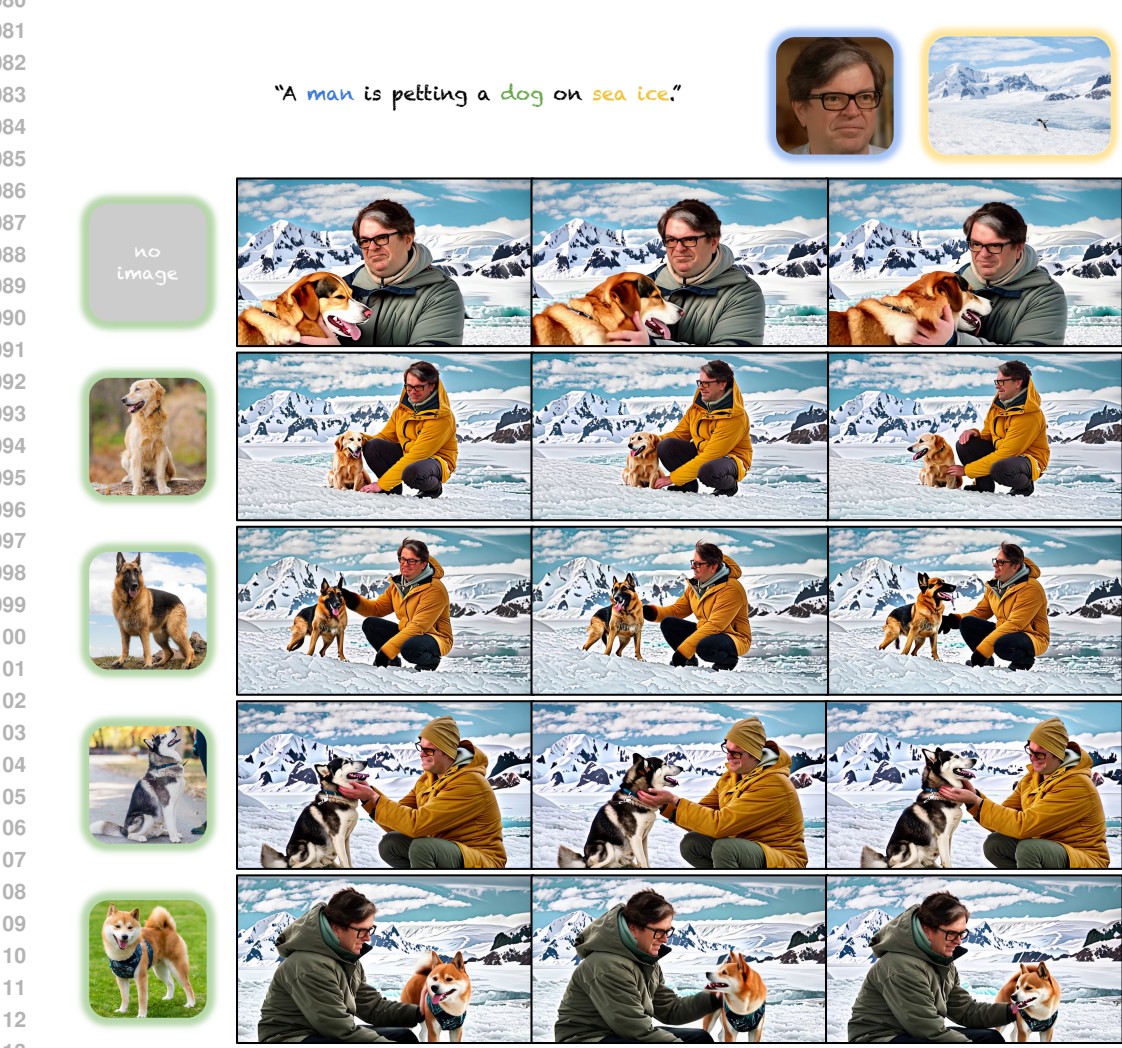

Figure 10: **Ablation study on the conditioning images of "*dog*".**

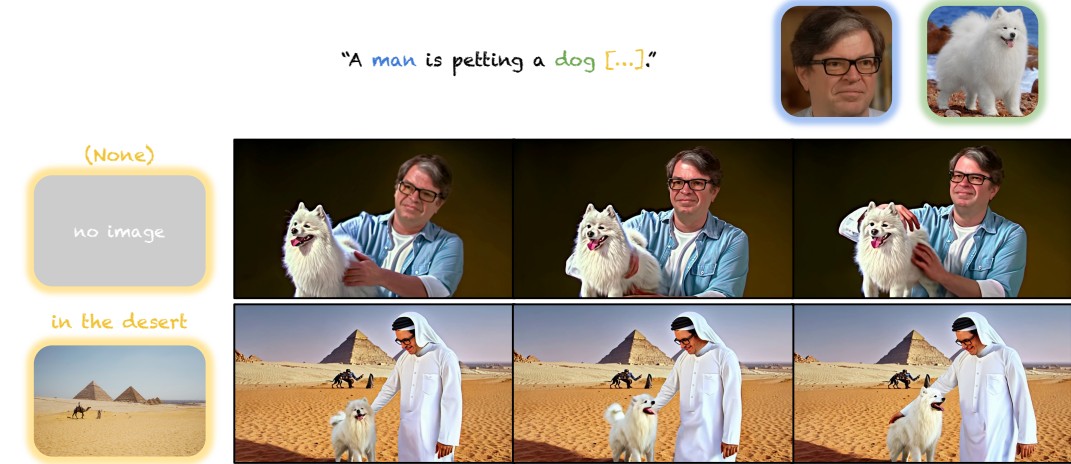

Figure 11: **Ablation study on the background conditioning images.**

## C.2 More Comparisons on Different Conditioning Subjects

Next, we present additional qualitative comparisons with the state-of-the-art video personalization frameworks. Figure 5 shows the comparison using the conditioning subject of "*a car*". Here we illustrate the comparisons using "*a cat*" in Figure 12 and "*a dog*" in Figure 13. We provide the video samples and more comparisons in the *webpage msrvtt* folder of the supplementary material.

"A gray cat with black stripes sits in a gray cat bed and yawns in a dimly lit room with a beige wall and a brown door."

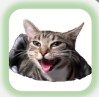

| ELITE | VideoBooth | Ours (with CLIP) | Ours (with DINOv2) |

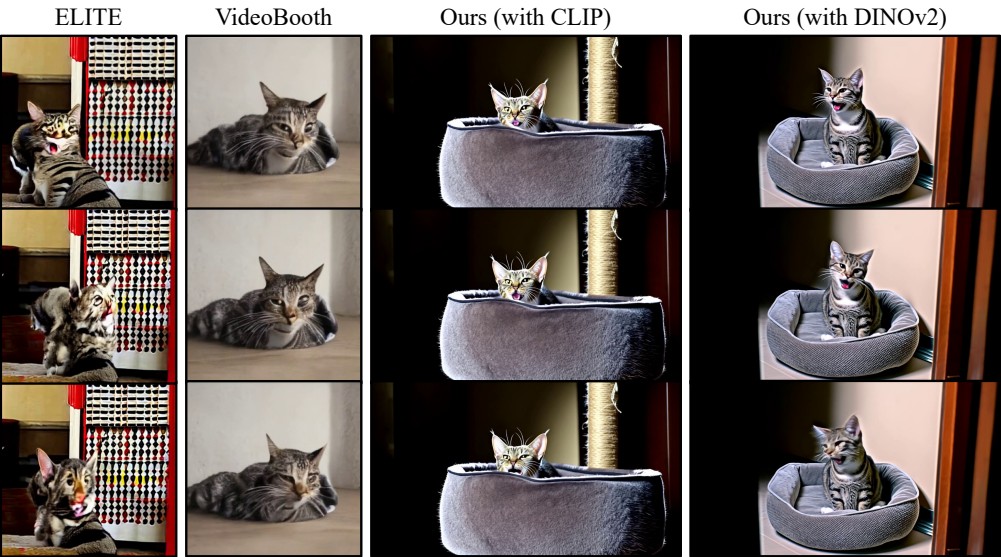

Figure 12: **Qualitative comparison on the conditioning subject of "a cat".**

"A brown and white puppy sits and stands up in a room with beige walls and a brown carpet and floor, next to a brown cabinet."

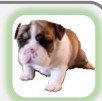

| ELITE | VideoBooth | Ours (with CLIP) | Ours (with DINOv2) |

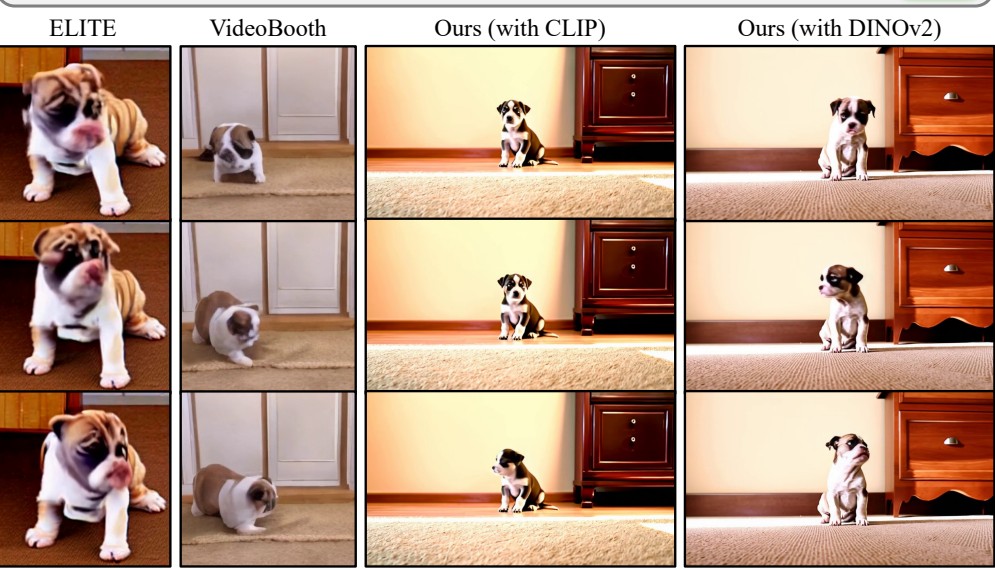

Figure 13: **Qualitative comparison on the conditioning subject of "a dog".**

# D    LIMITATIONS

**Undesirable Training Data Bias.** In Section 3.3, we address the issue of undesirable image biases by introducing augmentations and random sampling with replacement during training. However, some image biases, such as facial expressions and body postures, remain unresolved. As a result, our framework may generate subjects with similar facial expressions or postures as the reference images. Figures 12 and 13 show that existing personalization frameworks (Wei et al., 2023a; Jiang et al., 2024) with the same reconstruction-based learning strategy also exhibit this issue, which remains a challenge for future work.

**Taking Masked Images as Inputs.** Our model personalizes video synthesis using segmented reference image inputs. It requires users to provide masked images during inference and additional efforts may be needed if the localization algorithms do not segment the correct subject. Pasting the subject image segment to a random background image can be employed on the training dataset to address this issue.

**Oversaturation.** In Appendix B.3, we adopt classifier-free guidance (Ho & Salimans, 2022) (CFG) twice in each denoising step to achieve different CFG scales for text and personalization conditionings. However, we empirically observe that our model occasionally generates highly saturated samples, which is a persistent issue (Saharia et al., 2022; Kynkäänniemi et al., 2024) in diffusion models when strong CFG is used for sampling. For future work, we plan to explore sampling techniques like static or dynamic thresholding (Saharia et al., 2022) to address this issue.

**Unnatural Composition for Multiple Subject Conditioning.** When users input multiple subjects for conditioning, the synthetic videos sporadically exhibit unrealistic compositions and scales among the different subjects. This behavior can be interpreted as the relative minority of videos with multiple subjects in the training dataset. We are considering creating a training dataset featuring a higher frequency of video samples with multiple subjects for future work.

**Unsupported Measure on Video Quality.** Same as CLIP similarity score (Torchmetrics, 2024), *MSRVTT-Personalization* does not assess visual quality. Users must rely on alternative evaluations, such as user studies, to compare the visual quality.

