# OpenReview forum: "VideoAlchemy: Open-set Personalization in Video Generation"
_ICLR.cc/2025/Conference — ICLR 2025 Conference Withdrawn Submission_

### Official Review · Reviewer_5xgj · 2024-10-27

**Soundness:** 3
**Presentation:** 3
**Contribution:** 3
**Rating:** 3
**Confidence:** 5

**Summary:**

This paper introduces VideoAlchemy, a video customization method that generates personalized videos with multiple subjects and specified backgrounds, eliminating the need for test-time fine-tuning. The approach leverages a DiT architecture with an additional cross-attention layer and linear projection. Furthermore, this paper curates a large-scale training dataset, while employing data augmentation and conditional subject sampling strategy for training. For evaluation, this work develops a multi-subject video customization benchmark with four metrics. Experimental results demonstrate that VideoAlchemy surpasses existing methods both quantitatively and qualitatively.

**Strengths:**

1. This paper studies the multi-subject open-set video customization task, which is interesting and meaningful for practical video generation.
2. The paper is well-written and easy to understand.
3. This work curates a large multi-subject video dataset, containing 37.8M videos with subjects, objects, and backgrounds. It also introduces a multi-subject video customization evaluation benchmark.
4. The generated videos exhibit relatively high quality by using the DiT model and abundant data.

**Weaknesses:**

1. One of the important contributions of this paper is the curated dataset. However, I have concerns about:
    - The quality and diversity of the training and evaluation datasets. From Fig. 8's word cloud, humans and static objects/scenes (laptop, branch, house, sunset, building, etc.) make up a larger proportion of the dataset. Videos with static objects may have minimal motion. Also, based on my experience, the videos in MSRVTT primarily feature humans and scenes, with fewer animals and objects, potentially limiting test set diversity.
    - Availability of training dataset. While the paper promises to release the test dataset, it does not confirm making the training dataset publicly available. I believe releasing the training dataset is of great significance to the open-source community, especially since this paper uses the curated dataset as one of its main contributions.
2. The method's innovation is relatively limited. Although the differences from the IP-Adapter are mentioned in the method part, the similar idea of using separate cross-attention layers for text and image embeddings has been explored in previous work [1]. It would be better to discuss the differences.
3. The introduced evaluation benchmark mostly uses existing metrics and lacks comprehensiveness. It would be better to introduce new metrics, like temporal consistency or subject motion intensity as in VBench [2], to enhance the benchmark.
4. Another concern I have is about experiments:
    - Comparison Fairness. The method uses a DiT architecture, while comparison methods are mostly based on UNet, raising concerns that improvements might stem from a more robust video base model rather than the method itself.
    - Baseline Selection. It would be better to apply and reproduce image customization methods to video generative models, as in VideoBooth [3], rather than animation. On the other hand, there are some face customization works for video generation, such as Magic-Me[4] and ID-Animator [5]. It is best to compare with these methods to verify the effectiveness of the proposed method.
    - Lack of comparisons with multi-subject video customization methods. Since the paper studies the multi-subject customization task, existing methods like VideoDreamer [6], CustomVideo [7], and DisenStudio [8] should be discussed and chosen two or more for comparison.
    - Ablation Study. It would be better to conduct an ablation study on excluding text embeddings from image embeddings in image encoder.
5. I carefully viewed each video provided in the supplementary material. The video examples provided are not diverse enough for multi-subject video customization. There are only two categories, human and dog, and only one motion prompt "a ... is petting a dog on ...". It would be better to provide examples with different categories and prompts to demonstrate the effectiveness of the method on multi-subject customization.

[1] Li X, Hou X, Loy C C. When stylegan meets stable diffusion: a w+ adapter for personalized image generation.

[2] Huang Z, He Y, Yu J, et al. Vbench: Comprehensive benchmark suite for video generative models.

[3] Jiang Y, Wu T, Yang S, et al. VideoBooth: Diffusion-based video generation with image prompts.

[4] Ma Z, Zhou D, Yeh C H, et al. Magic-me: Identity-specific video customized diffusion.

[5] He X, Liu Q, Qian S, et al. ID-Animator: Zero-Shot Identity-Preserving Human Video Generation.

[6] Chen H, Wang X, Zeng G, et al. Videodreamer: Customized multi-subject text-to-video generation with disen-mix finetuning.

[7] Wang Z, Li A, Xie E, et al. Customvideo: Customizing text-to-video generation with multiple subjects.

[8] Chen H, Wang X, Zhang Y, et al. DisenStudio: Customized Multi-subject Text-to-Video Generation with Disentangled Spatial Control.

**Questions:**

1. Please refer to the weaknesses section. If my concerns are well addressed, I am willing to modify my score.
2. Typo: "issue" in line 60.
3. Typo: line 176 has two periods.
4. In line 339, the paper mentions using six metrics but only introduces four metrics.

---

> ### Author Response · Authors · 2024-11-14
> **Responses to Reviewer 5xgj (1/2)**
>
> We thank the reviewer for the insightful and valuable comments. Here are our responses:
>
> ---
>
> > "Limited quality and diversity of the training and evaluation datasets."
> >
> > "The video examples provided are not diverse enough for multi-subject video customization."
>
> To illustrate the diversity of our training dataset, we will demonstrate our model’s generalizability by including more synthetic videos with diverse conditional subjects (e.g., dog, cat, horse, car, dinosaur). We will also include more videos with multi-subject conditioning. In the meanwhile, we would like to emphasize that the word cloud is generated from a subset of the training dataset (10k samples out of a total of 37.8M videos) and thus does not fully represent the entire dataset's distribution.
>
> For the evaluation dataset, Figure 2 in the MSR-VTT paper confirms a diverse range of video content, covering topics such as sports, vehicles, and animals. Empirically, we observe that our testing dataset includes varied animals, including wolves, deer, and horses. We will visualize the qualitative comparison on these samples in the paper.
>
> ---
>
> > "Although the differences from the IP-Adapter are mentioned in the method part, the similar idea of using separate cross-attention layers for text and image embeddings has been explored in previous work"
>
> To equip VideoAlchemy with multi-subject, open-set personalization capabilities for both foreground objects and backgrounds, we design the model with two key innovations. One is separate cross-attention for image conditioning. The other is the binding of image and word concepts. To facilitate multi-image conditioning, it is necessary to provide the pairing between the conditional images and entity word tags. As such, we develop a novel algorithm to bind conditional images with entity word tags. We will include an ablation study to demonstrate the necessity of binding images with entity word tags.
>
> ---
>
> > "It would be better to introduce new metrics, like temporal consistency or subject motion intensity as in VBench, to enhance the benchmark."
> > "Baseline Selection."
>
> As suggested, we include *Dynamic Degree* and *Temporal Consistency* to evaluate the temporal quality. Additionally, we remove the application of the image animation algorithm and directly treat the outputs of the image models as single-frame videos for a fair evaluation. We also include the comparison with two optimization-based baselines: DreamVideo and Magic-Me.
> The results are shown as follows.
>
> **[Subject mode]**
> | Method  | Text-Sim | Video-Sim | Subject-Sim | Dync-Deg | Temp-Cons |
> | :-----: | :-----: | :-----: | :-----: | :-----: | :-----: |
> | `ELITE` | 0.245 | 0.620 | 0.359 | - | - |
> | `VideoBooth` | 0.222 | 0.612 | 0.395 | 0.448 | 0.963 |
> | `DreamVideo` | 0.261 | 0.611 | 0.310 | 0.311 | 0.956 |
> | `VideoAlchemy` | **0.269** | **0.732** | **0.617** | **0.466** | **0.993** |
>
> **[Face mode]**
> | Method  | Text-Sim | Video-Sim | Face-Sim | Dync-Deg | Temp-Cons |
> | :-----: | :-----: | :-----: | :-----: | :-----: | :-----: |
> | `IP-Adapter` | 0.251 | 0.648 | 0.269 | - | - |
> | `PhotoMaker` | **0.278** | 0.569 | 0.189 | - | - |
> | `Magic-Me` | 0.251 | 0.602 | 0.135 | 0.418 | 0.974 |
> | `VideoAlchemy` | 0.273 | **0.687** | **0.382** | **0.424** | **0.994** |
>
> Compared to the existing open-set personalization models, VideoAlchemy achieves notably higher fidelity while simultaneously having better text-aligned and temporally consistent videos with larger dynamics. More interestingly, our open-set model can also achieve significantly higher face fidelity than the models specific to the face domain.
>
> For the fairness of the comparison, the ablation study is conducted with the identical model architecture and the training data and can demonstrate the effectiveness of our proposed component.

---

> ### Author Response · Authors · 2024-11-14
> **Responses to Reviewer 5xgj (2/2)**
>
> ---
>
> > "Ablation Study."
> >
> > "Comparison Fairness."
>
> As suggested, we conduct an ablation study with three control factors. The results are shown as follows.
>
> | Method  | Image Encoder | Use Word Token | Image Augmentation | Text-Sim | Video-Sim | Face-Sim | Dync-Deg | Temp-Cons |
> | :-----: | :-----: | :-----: | :-----: | :-----: | :-----: | :-----: | :-----: | :-----: |
> | `Use CLIP` | CLIP | O | O | **0.269** | 0.768 | 0.569 | 0.552 | 0.990 |
> | `No word token` | DINOv2 | X | O | 0.256 | **0.790** | 0.566 | 0.569 | **0.991** |
> | `No augmentation` | DINOv2 | O | X | 0.251 | 0.781 | **0.609** | 0.506 | 0.987 |
> | `VideoAlchemy` | DINOv2 | O | O | 0.257 | **0.790** | 0.600 | **0.570** | 0.990 |
>
> **[Different Image Encoders]** We train the models with two encoders, CLIP and DINOv2, and find that the former yields better text similarity while the latter achieves better subject similarity. We hypothesize that DINOv2 is learned to minimize the self-supervised training objective and can capture unique features in an image. In contrast, CLIP is learned to bridge visual and textual modalities and focuses on details typically described in the prompt which can help the model generate text-aligned videos.
>
> **[Necessity of Binding Image and Word Concepts]** In Section 3.2, we bind word and image concepts by concatenating their embeddings. Empirically, we find that, without such binding, the model could apply image conditioning to an incorrect subject (e.g., put the input human face on a dog), which can lead to the missing of other subjects and lower subject similarity scores.
>
> **[Effect of Training Augmentations]** Section 3.3 introduces image augmentations during training to mitigate undesirable training data bias. Empirically, we find that training the model without augmentation can make it achieve better subject fidelity by copying and pasting the reference image into videos, which is not preferable and could lead to the degradation of the video dynamic.
>
> We will include both qualitative and quantitative comparisons in the paper.
>
> ---
>
> > "​​Lack of comparisons with multi-subject video customization methods."
>
> To the best of our knowledge, the existing multi-subject video customization methods, such as VideoDreamer, CustomVideo, and DisenStudio, do not release either the code or checkpoint. In addition, they are all arxiv papers at this moment.
>
> We also want to highlight that compared to multi-subject video customization, single-subject personalization is a relatively easier task and we have shown that VideoAlchemy surpasses the existing single-subject personalization model with a huge margin.

---

### Official Review · Reviewer_sPsG · 2024-10-30

**Soundness:** 2
**Presentation:** 3
**Contribution:** 3
**Rating:** 6
**Confidence:** 3

**Summary:**

The paper presents VideoAlchemy, a new video generation model capable of multi-subject, open-set personalization for both foreground and background elements. Built on a DiT architecture, it aims to handle personalization tasks across diverse subjects and settings. Different from those tuning-based methods, this paper direct trains the generation network from a large-scale dataset, eliminating the need for test-time optimization. To address the lack of paired video and image datasets, the authors introduce an automated data construction pipeline, alongside a new benchmark called MSRVTT-Personalization for evaluating subject fidelity in video synthesis.

**Strengths:**

● The model's design leverages separate cross-attention layers for text and image conditioning. It conceptually allows for effective handling of different modalities for more accurate conditioning. This design makes sense.

● This paper makes great efforts in building large-scale training datasets as well as the evaluation benchmark. If these can be publicly available, it will benefit research community.

● The paper proposes an effective data construction pipeline with extensive augmentations to counteract biases in training data.

● The model shows strong performance improvements in the proposed new task, particularly in preserving subject fidelity and achieving better video quality than previous methods.

**Weaknesses:**

Most importantly, these are a set of details missing in the paper, which will definitely harm its readability and contribution.

● For resource Requirements: The paper does not clarify the computational resources used for training (e.g., number of A100 GPUs, total GPU hours). As this model seems to be trained from scratch, this lack of detail makes it difficult to assess the model’s scalability for practical deployment​.

● For training data: Information about the training dataset's size, the number of training samples should be explicitly clarified in the main paper, rather than the supplementary file. Additionally, the details of how many training data samples are required to achieve the reported results are unclear.

● The benchmark and training set, which are claimed as key contributions, lack in-depth visualization and parameter analysis in the main text. Including charts that display data distribution, object categories, and scene types would make the evaluation more transparent​.

● The paper primarily compares numerical results and does not include generated video samples from other models, making it harder to visually assess the task's difficulty and the improvements made by the proposed model​.

● As claimed in the supplementary file, while the model supports longer video generation (16s with FPS 24), the paper does not explore whether identity consistency is maintained throughout the long-sequence video, as this is very challenging task.

● There is no clear statement about whether the model will be open-sourced or if a demo will be provided.

● For user input, it is unclear whether users need to manually map images and prompts for each subject during conditioning or if the model automatically aligns these components​.

**Questions:**

See the Weaknesses.

---

### Official Review · Reviewer_13tB · 2024-11-03

**Soundness:** 3
**Presentation:** 3
**Contribution:** 2
**Rating:** 5
**Confidence:** 5

**Summary:**

The paper presents VideoAlchemy, a novel model for open-set video personalization that enables multi-subject, optimization-free video generation. It utilizes a Diffusion Transformer to integrate text prompts with reference images, addressing challenges in data collection and model evaluation. The paper introduces a unique benchmark, MSRVTT-Personalization, for evaluating video personalization performance on both foreground and background elements, showing that VideoAlchemy outperforms existing methods.

**Strengths:**

- **A Strong Video Personalization Model**: The paper introduces an open-set video personalization model that supports robust multi-subject capabilities. By leveraging large-scale pre-training, the model enhances efficiency by eliminating the need for test-time fine-tuning, which is a notable advancement.
- **Comprehensive Dataset Curation and Augmentation**: The paper develops an extensive dataset curation and augmentation pipeline aimed at reducing biases in training data, which strengthens the quality and generalizability of the model's outputs.
- **Introduction of MSRVTT-Personalization Benchmark**: The MSRVTT-Personalization benchmark provides a nuanced and valuable framework for evaluating personalized video generation, contributing a useful tool for future research in this area.

**Weaknesses:**

- **Unclear Methodological Contribution of the Model**: While the dataset curation, augmentation pipeline, and benchmark are well-developed, the model itself primarily utilizes off-the-shelf techniques. It would be beneficial for the authors to clarify any specific methodological contributions of the model component.
- **Limited Baseline Comparisons**: The paper does not exhaustively compare its method against relevant video customization approaches, which could provide important context for the performance claims. Furthermore, differences in data, models, and training configurations across the compared baselines complicate a fair evaluation.
- **Lack of Ablation Studies**: The paper provides limited methodological ablation studies, particularly concerning the proposed training augmentation techniques and data. A more thorough exploration of these components would help clarify their individual contributions and effectiveness.

**Questions:**

Please refer to the weaknesses.

---

> ### Author Response · Authors · 2024-11-14
> **Responses to Reviewer 13tB (1/2)**
>
> We thank the reviewer for the insightful and valuable comments. Here are our responses:
>
> ---
>
> > "Lack of Ablation Studies"
>
> As suggested, we conduct an ablation study with three control factors. The results are shown as follows.
>
> | Method  | Image Encoder | Use Word Token | Image Augmentation | Text-Sim | Video-Sim | Face-Sim | Dync-Deg | Temp-Cons |
> | :-----: | :-----: | :-----: | :-----: | :-----: | :-----: | :-----: | :-----: | :-----: |
> | `Use CLIP` | CLIP | O | O | **0.269** | 0.768 | 0.569 | 0.552 | 0.990 |
> | `No word token` | DINOv2 | X | O | 0.256 | **0.790** | 0.566 | 0.569 | **0.991** |
> | `No augmentation` | DINOv2 | O | X | 0.251 | 0.781 | **0.609** | 0.506 | 0.987 |
> | `VideoAlchemy` | DINOv2 | O | O | 0.257 | **0.790** | 0.600 | **0.570** | 0.990 |
>
> **[Different Image Encoders]** We train the models with two encoders, CLIP and DINOv2, and find that the former yields better text similarity while the latter achieves better subject similarity. We hypothesize that DINOv2 is learned to minimize the self-supervised training objective and can capture unique features in an image. In contrast, CLIP is learned to bridge visual and textual modalities and focuses on details typically described in the prompt which can help the model generate text-aligned videos.
>
> **[Necessity of Binding Image and Word Concepts]** In Section 3.2, we bind word and image concepts by concatenating their embeddings. Empirically, we find that, without such binding, the model could apply image conditioning to an incorrect subject (e.g., put the reference human face on a dog), which can lead to the missing of other subjects and lower subject similarity scores.
>
> **[Effect of Training Augmentations]** Section 3.3 introduces image augmentation during training to mitigate undesirable training data bias. Empirically, we find that training the model without augmentation can make it achieve better subject fidelity by copying and pasting the reference image into videos, which is not preferable and could lead to the degradation of the video dynamic.
>
> We will include both qualitative and quantitative comparisons in the paper.
>
> ---
>
> > "Unclear Methodological Contribution of the Model"
>
> To equip VideoAlchemy with multi-subject, open-set personalization capabilities for both foreground objects and backgrounds, we design the model with several changes from the existing methods. We highlight two key innovations on the model design:
>
> **[Binding of Image and Word Concepts]** To facilitate multi-image conditioning, it is necessary to provide the pairing between the conditional images and entity words. As such, we develop a novel mechanism to bind the images with the corresponding entity words. We will include an ablation study to demonstrate the necessity of providing such binding information.
>
> **[Separate Cross-attention for Image Conditioning]** While IP-Adapter proposes a single decoupled cross-attention layer for both text and image conditioning, we empirically find that separate cross-attention layers work better for our case. We attribute this fact to two reasons:
> - First, multi-image conditioning introduces a longer sequence of image conditioning tokens. Therefore, mixing text and image tokens in a shared layer would lead to image tokens dominating the synthesis process and reducing text-video alignment.
> - Second, we apply T5 embeddings for text and DINOv2 embeddings for images. The mismatch of the embeddings distribution disables us to use a single cross-attention for both conditioning. Note that IP-Adapter uses CLIP embeddings for both conditional text and image. However, our experiments (see Tables 1 ~ 3 and Figures 5 ~ 6) reveal that DINOv2 image embeddings yield higher subject fidelity than CLIP.

---

> ### Author Response · Authors · 2024-11-14
> **Responses to Reviewer 13tB (2/2)**
>
> ---
>
> > "Limited Baseline Comparisons"
>
> As suggested, we additionally include the comparison with two optimization-based baselines: DreamVideo and Magic-Me. The results are shown as follows.
>
> **[Subject mode]**
> | Method  | Text-Sim | Video-Sim | Subject-Sim | Dync-Deg | Temp-Cons |
> | :-----: | :-----: | :-----: | :-----: | :-----: | :-----: |
> | `ELITE` | 0.245 | 0.620 | 0.359 | - | - |
> | `VideoBooth` | 0.222 | 0.612 | 0.395 | 0.448 | 0.963 |
> | `DreamVideo` | 0.261 | 0.611 | 0.310 | 0.311 | 0.956 |
> | `VideoAlchemy` | **0.269** | **0.732** | **0.617** | **0.466** | **0.993** |
>
> **[Face mode]**
> | Method  | Text-Sim | Video-Sim | Face-Sim | Dync-Deg | Temp-Cons |
> | :-----: | :-----: | :-----: | :-----: | :-----: | :-----: |
> | `IP-Adapter` | 0.251 | 0.648 | 0.269 | - | - |
> | `PhotoMaker` | **0.278** | 0.569 | 0.189 | - | - |
> | `Magic-Me` | 0.251 | 0.602 | 0.135 | 0.418 | 0.974 |
> | `VideoAlchemy` | 0.273 | **0.687** | **0.382** | **0.424** | **0.994** |
>
> Compared to the existing open-set personalization models, VideoAlchemy achieves notably higher fidelity while simultaneously having better text-aligned and temporally consistent videos with larger dynamics. More interestingly, our open-set model can also achieve significantly higher face fidelity than the models specific to the face domain.
>
> For the fairness of the comparison, we provide the ablation study above where the model architecture and the training data are identical, demonstrating the effectiveness of our proposed component.

---

### Official Review · Reviewer_PkRH · 2024-11-08

**Soundness:** 3
**Presentation:** 3
**Contribution:** 3
**Rating:** 5
**Confidence:** 3

**Summary:**

his paper proposes VideoAlchemy, synthesizing personalized videos with multiple subjects and open-set capabilities without time-consuming optimization. It integrates multiple conditions like reference images and text prompts. It proposes a an automatic data construction pipeline and a new personalization benchmark, in which the model outperforms existing methods.

**Strengths:**

1. The task of multi-subject open-set video customization is promising.
2. The paper is clearly written
3. This paper constructs a large-scale dataset and introduces a new personalization benchmark, which may promote future work.

**Weaknesses:**

1. The pipeline for dataset construction doesn't impress me much. Using Grounding DINO and SAM is quite common, as seen in [1,2]. Similar operations for face cropping are also demonstrated in [3].
2. The differences between the proposed method and IP-Adapter are minimal, and further discussion would be beneficial.
3. It seems that the paper does not mention plans to open-source the constructed large-scale dataset.
4. Temporal metrics were not used in the experiments, even though temporal consistency is another crucial factor in video alongside quality.
5. The experiments lack comparison with multi-subject video customization methods.

Reference
[1] VideoBooth: Diffusion-based video generation with image prompts.
[2] Boximator: Generating Rich and Controllable Motions for Video Synthesis.
[3] ID-Animator: Zero-Shot Identity-Preserving Human Video Generation.

**Questions:**

N/A

---

> ### Author Response · Authors · 2024-11-14
> **Responses to Reviewer PkRH (1/2)**
>
> We thank the reviewer for the insightful and valuable comments. Here are our responses:
>
> ---
>
> > "The pipeline for dataset construction doesn't impress me much. Using Grounding DINO, SAM, or face cropping are seen in VideoBooth, Boximator, and ID-Animator."
>
> Our goal is to build a video model equipped with multi-subject, open-set personalization capabilities for both foreground objects and backgrounds. This goal distinctly shapes our dataset collection pipeline and sets it apart from existing approaches. We highlight three key innovations:
>
> **[Multiple Entity Word Tags Extraction and Classification]** Different from previous methods, we utilize a LLM to extract multiple entity words from a text prompt and classify them into three categories: subject, object, and background. Additionally, we apply rigorous filtering criteria (as specified in lines 200 ~ 203 and Appendix A.2) to manage the increased complexity of retrieving multiple entity word tags within a single prompt.
>
> **[Learning Background Conditioning through Inpainting]** Our model uniquely supports background conditioning. To achieve this, we remove foreground elements by their dilated masks, and apply inpainting to create a clean background (as described in lines 211 ~ 215).
>
> **[Subject Image Extraction from Multiple Frames]** For each subject, we capture three images from distinct video frames, offering two advantages. First, it introduces variations in pose and lighting, addressing the *copy-and-paste* problem, a common issue of reconstruction-based personalization models. Second, this dataset enables the training with multi-image conditioning for individual subjects. Tables 1 and 2 demonstrate that multi-image conditioning can enhance subject fidelity in video synthesis.
>
> ---
>
> > "The differences between the proposed method and IP-Adapter are minimal, and further discussion would be beneficial."
>
> Compared to IP-Adapter, our approach incorporates three essential modifications to enhance high-quality multi-subject personalization:
>
> **[Binding of Image and Word Concepts]** To facilitate multi-image conditioning, it is necessary to provide the pairing between the conditional images and entity words. As such, we develop a novel mechanism to bind the images with the corresponding entity words. We will include an ablation study to demonstrate the necessity of providing such binding information.
>
> **[Application of Image Augmentation]** While both IP-Adapter and VideoAlchemy adopt a reconstruction-based training, we observe that this approach can lead to undesirable training data bias and a *copy-and-paste* issue, as described in Section 3.3. To mitigate this issue, we introduce image augmentations during training. We will provide an example illustrating the *copy-and-paste* limitation in IP-Adapter and include the ablation study to demonstrate the effectiveness of our proposed image augmentation strategy.
>
> **[Separate Cross-attention for Image Conditioning]** While IP-Adapter proposes a single decoupled cross-attention layer for both text and image conditioning, we empirically find that separate cross-attention layers work better for our case. We attribute this fact to two reasons:
> - First, multi-image conditioning introduces a longer sequence of image conditioning tokens. Therefore, mixing text and image tokens in a shared layer would lead to image tokens dominating the synthesis process and reducing text-video alignment.
> - Second, we apply T5 embeddings for text and DINOv2 embeddings for images. The mismatch of the embeddings distribution disables us to use a single cross-attention for both conditioning. Note that IP-Adapter uses CLIP embeddings for both conditional text and image. However, our experiments (see Tables 1 ~ 3 and Figures 5 ~ 6) reveal that DINOv2 image embeddings yield higher subject fidelity than CLIP.

---

> ### Author Response · Authors · 2024-11-14
> **Responses to Reviewer PkRH (2/2)**
>
> ---
>
> > "Temporal metrics were not used in the experiments, even though temporal consistency is another crucial factor in video alongside quality."
>
> As suggested, we include *Dynamic Degree* and *Temporal Consistency* to evaluate the temporal quality. The results are shown as follows.
>
> **[Subject mode]**
> | Method  | Text-Sim | Video-Sim | Subject-Sim | Dync-Deg | Temp-Cons |
> | :-----: | :-----: | :-----: | :-----: | :-----: | :-----: |
> | `ELITE` | 0.245 | 0.620 | 0.359 | - | - |
> | `VideoBooth` | 0.222 | 0.612 | 0.395 | 0.448 | 0.963 |
> | `DreamVideo` | 0.261 | 0.611 | 0.310 | 0.311 | 0.956 |
> | `VideoAlchemy` | **0.269** | **0.732** | **0.617** | **0.466** | **0.993** |
>
> **[Face mode]**
> | Method  | Text-Sim | Video-Sim | Face-Sim | Dync-Deg | Temp-Cons |
> | :-----: | :-----: | :-----: | :-----: | :-----: | :-----: |
> | `IP-Adapter` | 0.251 | 0.648 | 0.269 | - | - |
> | `PhotoMaker` | **0.278** | 0.569 | 0.189 | - | - |
> | `Magic-Me` | 0.251 | 0.602 | 0.135 | 0.418 | 0.974 |
> | `VideoAlchemy` | 0.273 | **0.687** | **0.382** | **0.424** | **0.994** |
>
> We show that VideoAlchemy can produce videos with text-aligned content, high-fidelity subjects, and appropriate video dynamics while maintaining temporal consistency.
>
> ---
>
> > "​​The experiments lack comparison with multi-subject video customization methods."
>
> To the best of our knowledge, the existing multi-subject video customization methods, such as VideoDreamer, CustomVideo, and DisenStudio, do not release either the code or checkpoint. In addition, they are all arxiv papers at this moment.
>
> We also want to highlight that compared to multi-subject video customization, single-subject personalization is a relatively easier task and we have shown that VideoAlchemy surpasses the existing single-subject personalization model with a huge margin.

---

### Note · Authors · 2024-11-15

I have read and agree with the venue's withdrawal policy on behalf of myself and my co-authors.